# Pore Water Chemical Variability and Its Effect on Phenological Production in Three Mangrove Species under Drought Conditions in Southeastern Mexico

**Claudia M. Agraz-Hernández [1], Carlos A. Chan-Keb [2,\*], Raquel Muñiz-Salazar [3], Román A. Pérez-Balan [2], Gregorio Posada Vanegas [1], Hector G. Manzanilla [1], Juan Osti-Sáenz [1] and Rodolfo del Río Rodríguez [1]**

[1] Instituto EPOMEX, Universidad Autónoma de Campeche, Av. Heroe de Nacozari #480, Campus 6 de Investigaciones, Campeche 24029, Mexico

[2] Facultad de Ciencias Químico Biológicas, Universidad Autónoma de Campeche, Avenida Ing. Humberto Lanz Cárdenas S/N, Colonia Ex Hacienda Kalá, Campeche 24085, Mexico

[3] Laboratorio de Epidemiología y Ecología Molecular, Escuela Ciencias de la Salud, Universidad Autónoma de Baja California, Blvd. Zertuche y Blvd. De Los Lagos S/N, Fracc. Valle Dorado, Mexicali 22890, Mexico

\* Correspondence: carachan@uacam.mx; Tel.: +52-981-8119800

**Abstract:** Mangrove forests have proven to be resilient to most environmental change, surviving catastrophic climate events over time. Our study aimed to evaluate the chemical variability of pore water and its effect on phenological production in three mangrove species (*Rhizophora mangle*, *Avicennia germinans*, and *Laguncularia racemosa*) along the coast of the state of Campeche during a year of severe drought (2009) and a year of average precipitation (2010). Pore water salinity and redox potential were measured monthly in a mangrove forest in 2009 and 2010. Litterfall production and reproductive phenology was measured monthly. We determined the relationships among litterfall production, reproductive phenology, pore water chemistry and precipitation of three species between years. Precipitation, pore water salinity and redox potential significantly differed among years, seasons and sites, and also showed significant interaction between years and seasons ($p < 0.05$). Significant variation was observed in litterfall production, propagules, flowers, and leaf litter among sites ($p < 0.05$). A significant change was observed in propagules and flowers among years, and in total litterfall and leaf litter between seasons and species ($p < 0.05$). Under severe drought, salinity had the strongest effect on total litterfall and propagules in *R. mangle*, while *A. germinans*, had the strongest effect on propagule/flower precipitation. Both *A. germinans* and *L. racemosa* showed higher resilience than *R. mangle* at all sites under severe drought conditions. These findings can support activity allocation for mangrove conservation and restoration by providing the tolerance thresholds of the three species that dominate in the regional area of Campeche state. Likewise, this research provides knowledge to the Intergovernmental Experts Group on climate change about drought intensity and its magnitude of impact on mangrove productivity, reproduction and integrity.

**Keywords:** mangrove; phenology; climate change; drought; vulnerability; litterfall; salinity stress

## 1. Introduction

Mangroves are considered key ecosystems internationally for maintaining coastal zone productivity; their conservation also facilitates the mitigation of and adaptation to climate change [1–3] due to their high capacity to capture and store atmospheric carbon in tree biomass and roots underground for long periods [4,5]. However, in the past 30 years, hydrometeorological events and anthropogenic changes have affected the environmental conditions in different types of forests in many coastal regions worldwide [2,6,7]. In addition, changes in rainfall, hydroperiod, and the physicochemical properties of pore water and sediments affect the functions and structure of mangrove communities which

has an impact on carbon sequestration [8,9]. Slight variations in precipitation regimes can lead to changes in composition, richness and productivity [10]. Gradients of pore-water salinity are caused by changes in precipitation, flooding regimes, microtopography and hydroperiod, and all of these factors affect the regulation of critical physiological processes in mangroves, such as photosynthesis [11]. A previous study on the Indian coast discovered significant inverse correlations between groundwater oxygen concentration and salinity and temperature [12]. The Intergovernmental Panel on Climate Change (IPCC) [13] has reported that a reduction in precipitation decreases the freshwater contribution, making the soil more saline. Consistent with this idea, [10] suggested that an increase in salinity can cause severe reductions in mangrove-covered surfaces due to the transformation of the upper tidal zones into hypersaline flats [14].

Mexico is particularly vulnerable to rainfall deficits (droughts) [15]. Most of the northern part of the country is naturally dry, whereas the southern part is wetter. However, sustaining heavy precipitation and containing water in natural underground reservoirs, lakes and rivers is also susceptible to severe drought effects. The Yucatán Peninsula, in particular, has experienced severe droughts [16,17]. With its extensive continental platform and high rainfall, the Gulf of Mexico promotes the development of extensive coastal wetlands [18], and these wetlands are widespread in the Yucatán Peninsula. Because of the climate variation in Campeche, there are alternating periods of low freshwater availability and high freshwater availability. This leads to soil salinization, sustained waterlogging, low oxygen levels, and low pH levels [19,20]. Agraz-Hernández et al. [21] demonstrated that significant changes in pore-water chemical conditions and the production of total litterfall and propagules were shown to be driven by annual variation in seasonal precipitation between 2006 and 2010. These changes occurred in monospecific forests of *Rhizophora mangle* L. located in Laguna de Terminos, Campeche.

This study aimed to compare variations in hydrochemical characteristics of interstitial water, rainfall, phenological and litterfall production of the three mangrove species (*Rhizophora mangle*, *Avicennia germinans* and *Laguncularia racemosa*) from the coast of Campeche in a severe drought year (2009) against average rainfall year (2010). Specifically, we asked the following questions: (1) Are there significant variations in salinity, pore water redox potential and rainfall when comparing a drought year against a rainy year according to sites located on the coast of Campeche? (2) What is the relationship between litterfall production and phenology of mangrove species with hydrochemistry and rainfall on study sites along the Campeche coast? Finally, (3) which species showed the best adaptation to drought conditions?

## 2. Materials and Methods

### 2.1. Study Area

The state of Campeche is located in southern Mexico and has 523 km of coastline along the Gulf of Mexico [22]. It has a tropical savanna climate, which is classified as Aw by the Köppen–Geiger system, and experiences heavy rainfall during the summer from June to October [23,24]. Precipitation increases from the southeast (1400 mm year$^{-1}$) to the northwest (600 mm year$^{-1}$) [22]. Campeche has the largest area of mangroves on the Mexican coast (25.6%) and the best-preserved forests, with a total area of 200,279 ha [25]; it is thus considered one of the most important coastal ecosystems in Mesoamerica [26]. The most extensive area of mangroves (107,262 ha) [14] is located at Laguna de Terminos (LT) in southwestern Campeche, and the most conserved mangrove forest is located at Los Petenes Biosphere Reserve (RBP) in northwestern Campeche (73,776 ha) [14]. The dominant species in the southwest is *A. germinans* and *R. mangle* is the dominant species in the northwest. In this study, we analyzed seven mangrove forest sites comprising the southeastern and northeastern parts of the state of Campeche and divided into three zones: the *Laguna de Términos* (LT; a) Protected Natural Area, with four sampling sites (Atasta, Estero Pargo, Sabancuy and Xibuja); *Río Champotón* (RC; b), with one site; and *Los Petenes*



*Biosphere Reserve* (RBP; c), with two sites (Río Verde and Peten Neyac) (Figure 1). These sites were selected through a previous survey of 400 km of the study area [1,14] and the Google Earth viewer. Variations were identified in the number and diameter of trees, dominant species and environmental indicators (i.e., flood amplitude, stress by non-native sources to the ecosystem among others).

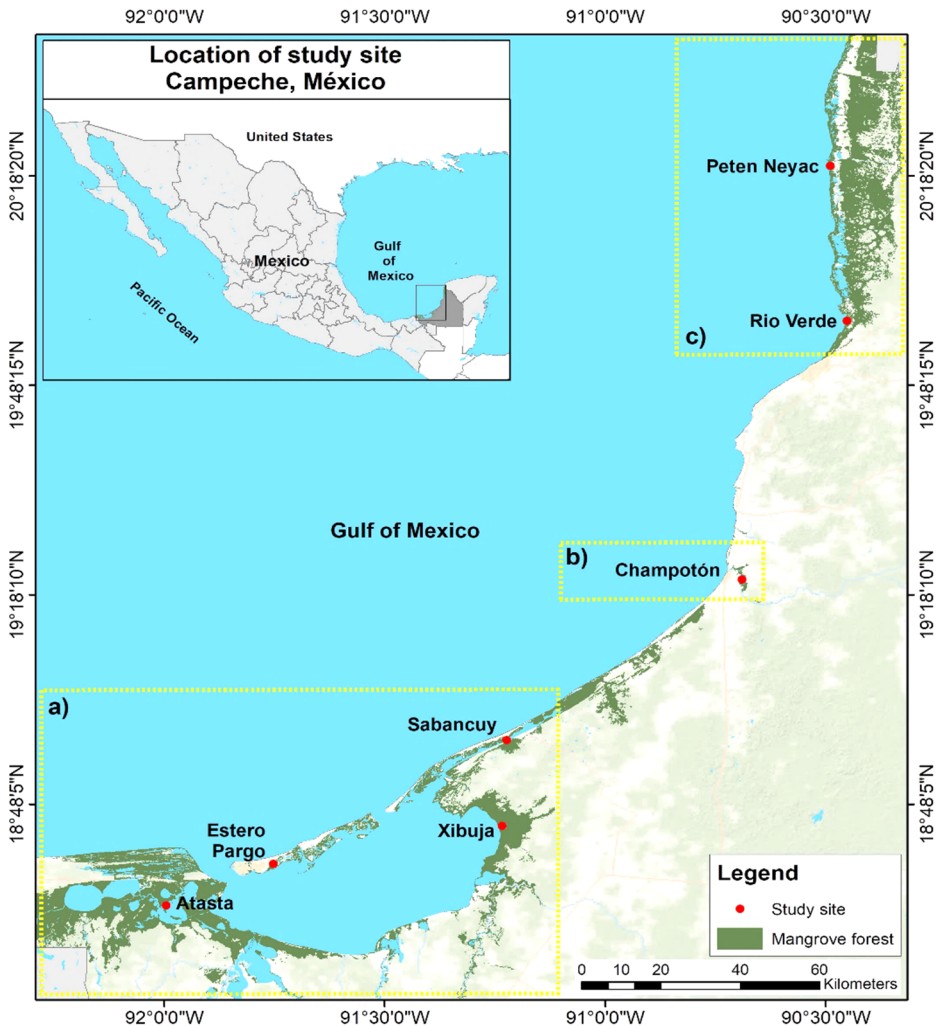

**Figure 1.** Map of the seven mangrove forests studied on the coast of Campeche, Mexico. (**a**) Laguna de Terminos, (**b**) Rio Champoton and (**c**) Los Petenes Biosphere Reserve.

### 2.2. Analysis of Drought

The standardized precipitation index (SPI) was used to assess deficits and excesses in precipitation in 25 years (between 1995 and 2020) and define a reasonable time period to establish the expected rainfall during the lifetime of the mangroves. Likewise, this represents precipitation evolution from 1995 to 2020 in the climatological stations nearest to the sampling sites of Laguna de Terminos (No. 4015), Champoton (4008–4041) and Los Petenes Biosphere Reserve (4038). The SPI evaluates deficits and excesses in precipitation for a site during a given period of one month to up to four years. The SPI is computed by dividing the difference between the normalized seasonal precipitation and its long-term seasonal mean by the standard deviation [27,28]:

$$\text{SPI} = \frac{X_{ij} - X_{im}}{\sigma} \tag{1}$$

where $X_{ij}$ is the seasonal precipitation at the $i^{th}$ rain gauge station and $j^{th}$ observation, $X_{im}$ the long-term seasonal mean and $\sigma$ is its standard deviation.

Since the SPI is equal to the z-value of the normal distribution, McKee et al. [29] proposed a seven-category classification for the SPI (Table 1).

The expected time in each drought category was based on an analysis of many rainfall stations across Colorado, USA. The percentage of time spent in moderate, severe and extreme drought corresponds to those expected from a normal distribution of the SPI [30].

**Table 1.** Standardized precipitation index (SPI) adapted from McKee et al. [29] and Chan Keb et al. [14].

| Category | SPI |
|---|---|
| Extreme wet | ≥2.0 |
| Severe wet | 1.50–1.99 |
| Moderate wet | 1.00–1.49 |
| Humidity | 0.50–0.99 |
| Normal prone to wet | 0.00–0.49 |
| Normal prone to dry | −0.00 to −0.49 |
| Drought | −0.50 to −0.99 |
| Moderate Drought | −1 to −1.49 |
| Severe Drought | −1.50 to −1.99 |
| Extreme Drought | ≤2.0 |

Seasonal behavior was established with rainfall and temperature records (from 1995 to 2020 year) of meteorological stations near interest sites in the study area. The seasonal differences between the 2019 and 2020 years were analyzed through a multivariate cluster analysis using the Ward link method and Euclidean distance.

### 2.3. Wind

Through the Copernicus ERA 5 database (2021) [31] for the years 2009–2010, the behavior of the wind at a 10 m height was obtained for the coordinates Lat 18.75–Lon −91.75, Lat 19.50–Lon −90.75 and Lat 20.25–Lon −90.75, which are representative of Laguna de Terminos, Río Champotón and Los Petenes, Campeche, respectively.

### 2.4. Forest Attributes and Physiognomic Type

The mangrove forest attribute measurements (diameter, height, density and basal area) were carried out in January 2019. We used two 10 × 10 m (100 m²) quadrats in each site with *A. germinans*, *L. racemosa* and *R. mangle*, and each quadrat was laid perpendicular to the shoreline. Quadrats were selected to represent all the species in that area. Each species in the quadrant was counted. The tree diameter of *A. germinans* and *L. racemosa* was determined at 130 cm above ground and that for *R. mangle* was determined at 30 cm above the highest prop-root [32] with diameter greater than 1 cm with a tape *pi* using a SUUNTO clinometer and a penta decameter. In forests where mangrove trees had a stem diameter ≤2.5 cm, quadrants of 5 × 5 m were used. Forest attributes were measured by the formulas shown in Table S1 [33,34].

The physiognomic type of the forest was determined through the density, basal area and height of the forest following reference [35].

### 2.5. Reproductive Phenology Based on Litterfall

Litterfall production was measured each month based on the dry weight of total litterfall and its components for *R. mangle*, *A. germinans* and *L. racemosa* collected in 2009 and 2010. In each forest, 14 litterfall baskets (0.50 × 0.50 m) made of 1 mm mesh and that were 30 ± 15 cm deep were installed, within an area of 0.1 ha, following a random spatial pattern according to the presence of mangroves species within each monitoring location [36], with a total of 98 litterfall baskets. The baskets were suspended 1.5 m above the highest tide

level and storm tide level under the tree canopy using nylon threads tied to each basket corner. The collected material was dried in a convection oven at 65 °C for approximately three days until constant weight was reached [36,37]. The dry matter was separated into leaves, flowers, propagules, branches, stipules (basal appendix of a petiole) and miscellaneous (material with a high degree of degradation without being able to be identified) [38] and was weighed using a digital scale with 0.001 g precision (Precisa brand). We grouped the data into monthly sets of seasonal and yearly groups to conduct the statistical analyses

We calculated productivity as the total litterfall dry weight and as the proportion of each type of litterfall dry weight per square meter per month and per year.

### 2.6. Pore Water Chemistry

The mangrove forest pore water chemistry (salinity and redox potential) was measured in situ each month from February 2009 to December 2010 using piezometers. Pore water is defined as the water that flows between sediment particles. The redox potential measures the activity of the electrons of the chemicals found in the ground water, and it is related to the oxygen dissolved in water [39]. We installed three polyvinyl chloride (PVC) piezometers of 10.16 cm in diameter and 1.5 m in length at a depth of 0.50 m in the sediment surface of each mangrove forest. Most of the tree root biomass is usually located at this depth [36,40]. The tubes were installed each litterfall plot of 10 × 10 m. The first tube was placed at the edge of the shoreline and the second and third tubes were installed inside the mangrove forest and separated by 5 m. These 1 cm tube holes were drilled to a depth of 30 cm. Water samples were collected after the tubes were drained and the water was stabilized [41]. Redox potential was measured using a HACH HQ40d portable multimeter (Loveland, CO, USA). The HQd meter had an oxide reduction (redox) potential (ORP) probe that was not gel-filled. The ORP solution used intervals of ±1200 mV. Salinity was determined with refractometers (ATAGO, Inc., Bellevue, WA, USA) with a measurement range of 0 to 100 practical salinity units (PSU) [36,40]. Salinity and redox potential classification were performed according to Cronk et al. [42].

### 2.7. Statistical Analysis

The pore-water chemistry variability and phenological characteristics between sites (seven sites), years (2009 and 2010), seasons (dry, Nortes and rainy) and species (*R. mangle*, *A. germinans* and *L. racemosa*) were compared using a 7 × 2 × 3 × 3 factorial (sites, years, season and species) analysis of variance (ANOVA) Subsequently, Fisher's least significant difference test (LSD) was applied post hoc to determine significant differences between means using an $\alpha$ = 0.05 significance level. The normal distribution of data was analyzed using the Shapiro–Wilk test with an $\alpha$ = 0.05 significance level; data that did not show a normal distribution were transformed using the Box–Cox method as recommended in Zar [43]. We used Pearson correlation analysis to examine the correlations between precipitation and years, salinity, redox potential and litterfall. Canonical correlation analysis was used to characterize the relationships among litterfall production, reproductive phenology, pore-water chemistry variability and precipitation of the three mangrove species under two different scenarios: severe drought (2009) and high humidity conditions (2010). All statistical analyses were performed using Statistica V.12 (StatSoft, Inc., Tulsa, OK, USA, 1984–2014).

### 3. Results

### 3.1. Drought Analysis

Fifty-three coastal stations (located less than 50 km from the coastline) were selected from 88 CONAGUA meteorological stations belonging to the State of Campeche CLICOM database. With these data, the monthly variation in precipitation from 1959 to 2010 was determined.

Figure 2 shows the average values, 95% confidence intervals and average annual precipitation (1216.3 ± 110 mm). According to these data, September is the rainiest month (219.00 ± 40 mm), and March is the least rainy month (21.12 ± 10 mm).

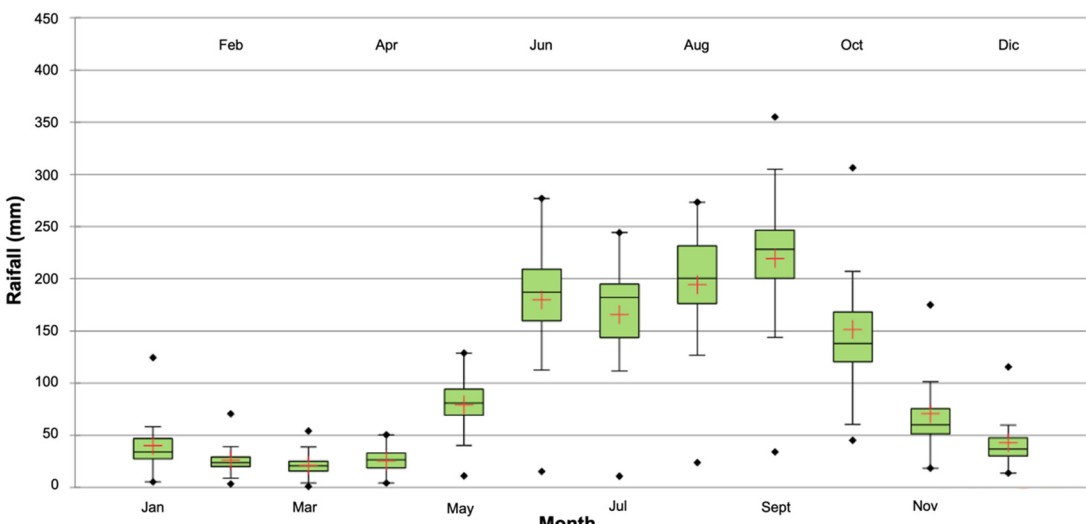

**Figure 2.** Shows the average values, 95% confidence intervals and average annual precipitation (1216.3 ± 110 mm). According to these data, September is the rainiest month (219.00 ± 40 mm), and March is the least rainy month (21.12 ± 10 mm), squares represent minimum and maximum values for each month.

The seasonal distribution of precipitation three different seasons a year throughout the coastal littoral of the State of Campeche: (a) minimum precipitation (dry season) from February to April, (b) maximum precipitation (rainy season) from May to October and (c) intermediate precipitation with wind from the Gulf of Mexico ("Nortes" season) from November to January (Figure 3). These results are similar to those reported by Barreiro-Güemes [23] and INEGI [24].

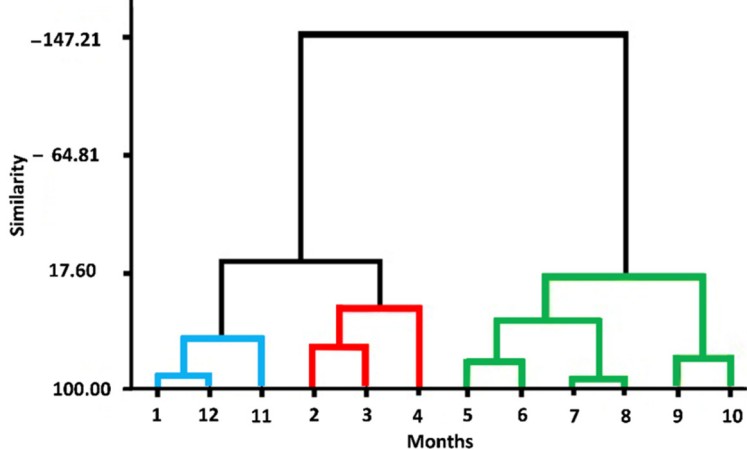

**Figure 3.** Cluster analysis for monthly precipitation and temperature in the period 1959–2010 recorded at meteorological stations near seven study sites. The months were grouped into three seasons of the year: blue lines correspond to the Nortes season, red lines refer to the dry season and green to the rainy season.

Along the coast of Campeche, the precipitation was heterogeneous between 1995 and 2020 (Figure 4). Droughts were registered in 1997, 2001 and 2011 (SPI values = −0.50, −0.58

and −0.64, respectively), and there was severe drought in 2009 (SPI values = −1.01). It should be noted that in these years a precipitation deficit was exhibited in the rainy season, including the years 1998 and 2004 in which drought conditions were recorded (−1.5 to −0.58 SPI). For the years 2009 and 2011, the dry season defined drought conditions (SPI values = −0.95 to −0.87 SPI). These scenarios establish the displacement of seasons throughout the year in the cited years (Figure 4).

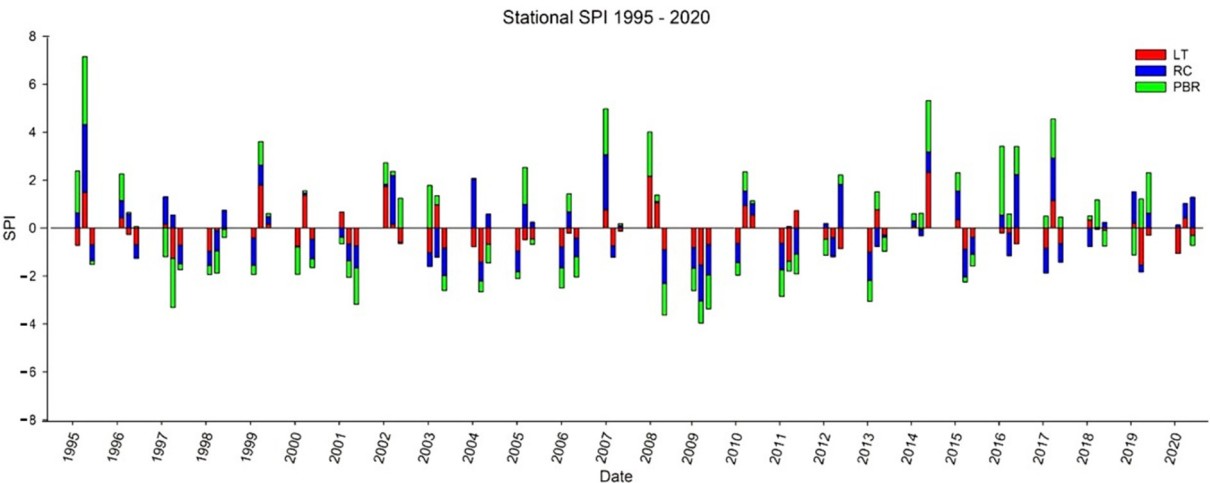

**Figure 4.** Standardized Precipitation Index (SPI) for the period of 1995 to 2020. SPI was calculated based on the annual precipitation data for 53 stations along the coasts of Campeche.

Total annual precipitation differed significantly between sites (F = 11.25, $p < 0.0001$), years (F = 8.86, $p < 0.003$) and seasons (F = 290.21, $p < 0.0001$) and a significant year–season interaction was detected (F = 32.31, $p < 0.0001$). Precipitation significantly differed between sites (LT vs. RBP and LT vs. CR) and between seasons in 2009 and 2010 ($p < 0.05$; Figure 5) (Table 2).

**Table 2.** Results of a three-factorial repeated-measurement ANOVA on the effects of site (Peten Neyac, Río Verde, Champotón, Sabancuy, Xibuja, Estero Pargo and Atasta), year (2009 and 2010), seasons (dry, rainy and Nortes) and species (*Rhizophora mangle*, *Avicennia germinans* and *Laguncularia racemosa*) on the precipitation along the Campeche coast, with a significance level of $p < 0.05$.

| Factor | Precipitation | | |
|---|---|---|---|
| | df | F | p |
| A (Site) | 6 | 11.25 | <0.0001 ** |
| B (Year) | 1 | 8.86 | <0.003 * |
| C (Season) | 2 | 290.21 | <0.0001 ** |
| A × B | 6 | 0.61 | 0.725 |
| A × C | 12 | 1.68 | 0.069 |
| B × C | 2 | 32.31 | <0.0001 ** |
| A × B × C | 12 | 0.60 | 0.839 |
| Error | 360 | | |

Significance level: $p < 0.05$ *, $p < 0.0001$ **.

During the rainy and Nortes seasons in 2009, moderate drought occurred (SPI = −1.12 and −1.32, respectively, Figure 5). During the dry season, drought conditions were detected (SPI = −0.87) (Figure 5). Specifically, severe drought was observed at LT during the rainy and Nortes seasons (SPI = −1.53 to −1.39, Figure 5). In 2010, wet conditions were

observed in the Nortes and rainy seasons (SPI = 0.52 to 0.78). However, during the dry season, only drought was observed at LT (SPI = −0.65), similar to 2009 (Figure 5).

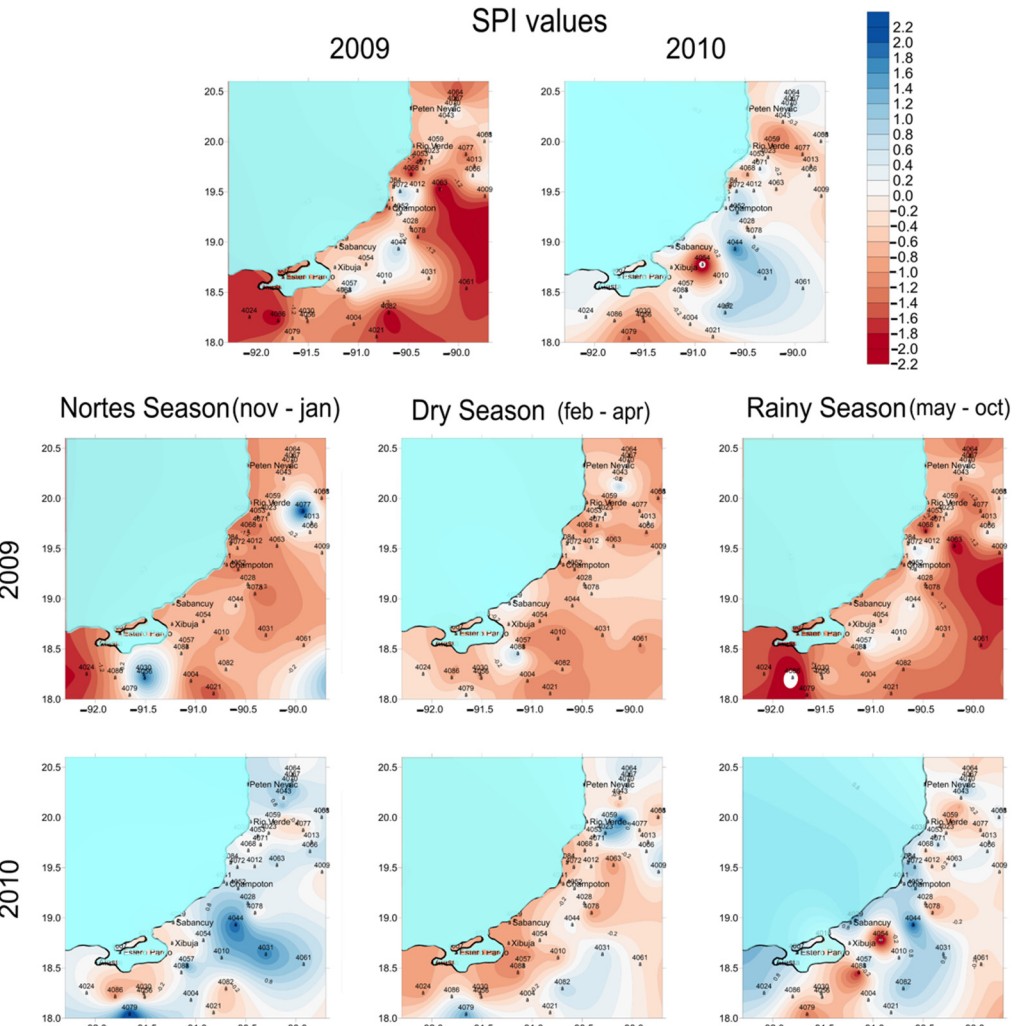

**Figure 5.** Standardized precipitation index (SPI) at three different times during 2009 and 2010.

### 3.2. Wind Analysis

For LT, RC and LP during 2009–2010. the maximum speeds registered are 8.98, 19.76 and 16.11 m/s corresponding to the months of February and June 2010, respectively. For these same months, the average speeds were 3.21, 5.82 and 5.43 m/s.

### 3.3. Forest Attributes and Physiognomy of Mangroves

Differences in forest physiognomy and species dominance were recorded along the coast of Campeche state (Table 3). The forest attributes from the southwest (LT) to northwest (RBP) decreased, the Atasta area was defined with full riparian physiognomy, and the rest of the sites in the LT also showed tendencies to riparian physiognomy (Table 3). The dominant species on the LT sites is *A. germinans*, with the presence of *R. mangle* and *L. racemosa*, except for Estero Pargo site, which had a dominance of *R. mangle* followed by *A. germinans* and *L. racemosa* (Table 3).

In the case of Champoton River forest, it exhibited basin physiognomy with a riparian tendency and dominance of *R. mangle*, with the presence of *A. germinans* and *L. racemosa* (Table 3).

The RBP forests registered an edge physiognomy, with a dominance of *R. mangle*, followed by *A. germinans* and *L. racemosa* (Table 3).

**Table 3.** Structural characteristics of mangrove forests of Campeche state, Mexico.

| Ecosystem | Sites | Total Density Trees (Stems ha⁻¹) | Total Basal Area (m² ha⁻¹) | Stand Height (m) | Species | Physonomic Type |
|---|---|---|---|---|---|---|
| Laguna de Terminos | Atasta | 1914.4 | 65.0 | 12.0 | Ag *, Rm, Lr | R |
| | Estero Pargo | 2562.7 | 20.1 | 6.1 | Rm *, Ag, Lr | R-F |
| | Sabancuy | 4784.2 | 15.5 | 6.0 | Ag *, Rm, Lr | R-F |
| | Xi buja | 3470.9 | 39.6 | 6.0 | Ag * | R-F |
| Rio Champoton | Champoton | 1090.0 | 5.8 | 7.9 | Rm *, Ag, Lr | B-F |
| Los Petenes Biosphere Reserve | Peten Neyac | 2930.8 | 25.3 | 7.3 | Rm *, Ag, Lr | F |
| | Río Verde | 2672.4 | 10.2 | 10.3 | Rm *, Ag, Lr | F |

R—River; R-F—River-fringer; F—fringer; B—Basin, Rm—*Rhizophora mangle*; Ag—*Avicennia germinans*; Lr—*Laguncularia racemosa*. * Dominant species.

*3.4. Temporal and Spatial Variation of Litterfall*

Litterfall (1320.8 g·m⁻²·year⁻¹) was, significantly, 14.2% lower in 2009 than 2010 (1538.8 g·m⁻²·year⁻¹) (Figure 6); (F = 41.31, $p < 0.0001$) (Table 4). Litterfall was lower in 2009 than in 2010 in each season: 17.4% lower during the rainy season, 7.8% lower during the Nortes season and only 1.6% lower during the dry season (Figure 6).

Factorial ANOVA showed that total litterfall varied significantly between seasons (F = 67.87, $p < 0.0001$). The maximum litterfall was recorded during the rainy season at all sites (Figure 6).

Litterfall was 18.12% lower in LT and 15.8% lower in RBP in 2009 than in 2010 (Figure 6). Factorial ANOVA showed that total litterfall varied significantly between sites (F = 6.15, $p < 0.0001$). Atasta in LT experienced a large decrease in total litterfall (35.8%), which coincides with the rain deficit in 2009 (Figure 6). Fisher's LSD post-hoc tests revealed significant differences in litterfall and pore-water redox potential between sites between 2009 and 2010 ($p < 0.05$) (Table S2). Factorial ANOVA showed that total litterfall varied significantly between species (F = 87.34, $p < 0.0001$) (Table 4; Figure 6).

**Table 4.** Four factorial repeated measurement ANOVAs on the effects of site (Peten Neyac, Río Verde, Champotón, Sabancuy, Xibuja, Estero Pargo and Atasta), year (2009 and 2010), seasons (dry, rainy and Nortes) and species (*Rhizophora mangle*, *Avicennia germinans* and *Laguncularia racemosa*) on the litterfall production and components along the Campeche coast, with a significance level of $p < 0.05$.

| | Litter Fall | | |
|---|---|---|---|
| **Factor** | **df** | **F** | ***p*** |
| A (Sites) | 6 | 6.15 | <0.0001 ** |
| B (Year) | 1 | 41.31 | <0.0001 ** |
| C (Season) | 2 | 67.87 | <0.0001 ** |
| D (Species) | 2 | 87.34 | <0.0001 ** |
| A × B | 6 | 0.29 | 0.943 |
| A × C | 12 | 0.879 | 0.576 |
| A × D | 12 | 15.32 | <0.0001 ** |
| B × C | 2 | 33.19 | <0.0001 ** |
| B × D | 2 | 2.50 | <0.043 * |
| C × D | 4 | 1.44 | 0.239 |
| A × B × C | 12 | 1.08 | 0.379 |
| A × B × D | 12 | 0.41 | 0.995 |
| A × C × D | 24 | 1.70 | 0.150 |

| | | | |
|---|---|---|---|
| B × C × D | 4 | 0.38 | 0.971 |
| A × B × C × D | 24 | 0.69 | 0.865 |
| Error | 293 | | |

Significance level: $p < 0.05$ *, $p < 0.0001$ **.

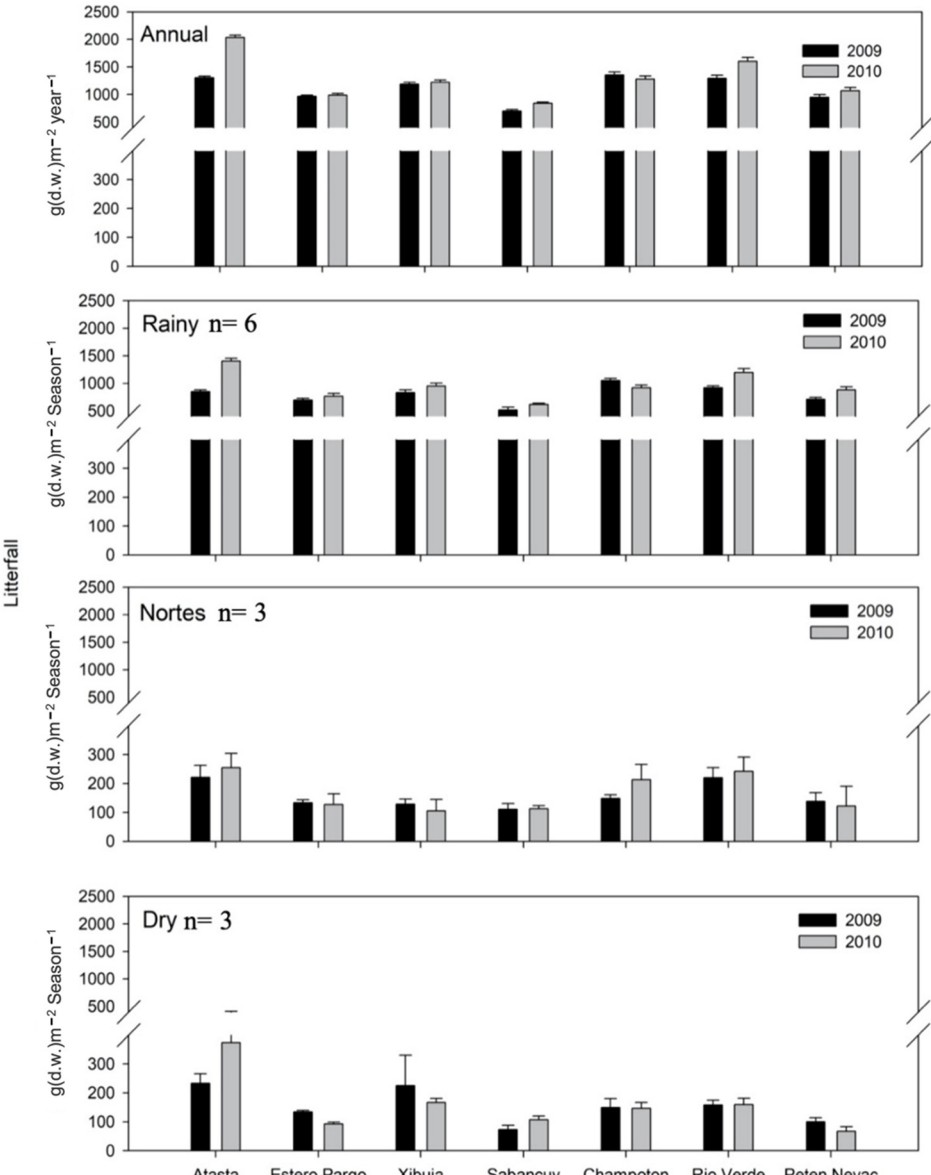

**Figure 6.** Litterfall (±SE = standard error) by year and season in seven mangrove forests along the coast of Campeche, Mexico for 2009 and 2010.

*3.5. Reproductive Phenology Based on Litterfall*

The production of leaf-litter varied significantly by season in 2009 and 2010 (F = 23.97, $p < 0.001$; Table 5); it was 7.9% higher during the rainy season, 6.2% higher during the dry season and 19.8% higher during the Nortes season; the increase in the rainy season was mainly due to *A. germinans*.

Leaf-litter was significantly different between sites ($p < 0.05$; Table 5). Higher leaf-litter was observed in LT during the rainy season in 2009 and 2010 compared with RBP, where flower fall significantly varied among sites depending on the season (F = 1.89, $p < 0.035$) (Table 5, Figure 7). Flower fall was 25% lower in 2009 than in 2010 (F = 13.06, $p <$

0.0001; Table 5), especially during the Nortes season, when it was 82.5% lower compared with the dry and rainy seasons (F = 31.04, *p* < 0.0001; Table 5).

Flower fall significantly differed between years and species (F = 6.75, *p* < 0.0001) (Table 5, Figure 7). Additionally, differences in flower fall between CR vs. Peten Neyac and Río Verde in RBP and vs. Estero Pargo in LT were significant (*p* < 0.05) (Table S2). Flower fall was highest for *R. mangle* in CR at RBP in 2009 (Figure 7).

Propagule fall significantly differed between years and seasons (F = 32.73, *p* < 0.0001, respectively) (Table 5). Between 2009 and 2010, a 56.7% decrease in propagule fall was observed. In 2009, propagule fall in the rainy and dry seasons was 60.7% and 67.7% lower than that in 2010, respectively. The difference between species was 47.5% lower in 2009 than in 2010 (F = 4.8, *p* < 0.001), depending on the season (F = 3.67, *p* < 0.027; Tables 6 and 7). Additionally, leaf (*p* < 0.027), flower (*p* < 0.001) and propagule (*p* < 0.001) production significantly differed (*p* < 0.05) between mangrove species depending on the season (Table 5).

**Table 5.** Results of a four-factorial repeated-measurement ANOVA on the effects of site (Peten Neyac, Río Verde, Champotón, Sabancuy, Xibuja, Estero Pargo and Atasta), year (2009 and 2010), season (dry, rainy and Nortes) and species (*Rhizophora mangle*, *Avicennia germinans* and *Laguncularia racemosa*) on the litter fall production and components along the Campeche coast, with a significance level of *p* < 0.05.

| Factor | df | Leaves | | Flowers | | Propagules | |
|---|---|---|---|---|---|---|---|
| | | F | *p* | F | *p* | F | *p* |
| A (Sites) | 6 | 7.30 | <0.0001 | 7.36 | <0.0001 ** | 6.26 | <0.0001 ** |
| B (Year) | 1 | 1.30 | 0.256 | 13.06 | <0.0001 ** | 0.01 | 0.926 |
| C (Season) | 2 | 23.97 | <0.0001 | 31.04 | <0.0001 ** | 32.74 | <0.0001 ** |
| D (Species) | 2 | 91.52 | <0.0001 | 199.18 | <0.0001 ** | 36.39 | <0.0001 ** |
| A × B | 6 | 0.98 | 0.438 | 1.26 | 0.28 | 0.46 | 0.835 |
| A × C | 12 | 1.48 | 0.129 | 1.89 | <0.035 * | 1.18 | 0.298 |
| A × D | 12 | 17.73 | <0.0001 | 9.28 | <0.0001 ** | 10.64 | <0.0001 ** |
| B × C | 2 | 0.38 | 0.681 | 0.05 | 0.95 | 0.85 | 0.429 |
| B × D | 2 | 1.71 | 0.148 | 6.75 | <0.0001 ** | 4.80 | <0.001 * |
| C × D | 4 | 0.75 | 0.472 | 1.30 | 0.27 | 3.67 | <0.027 * |
| A × B × C | 12 | 0.85 | 0.604 | 0.67 | 0.79 | 1.08 | 0.376 |
| A × B × D | 12 | 0.43 | 0.992 | 0.97 | 0.51 | 0.57 | 0.952 |
| A × C × D | 24 | 0.78 | 0.539 | 2.27 | <0.049 * | 1.05 | <0.383 |
| B × C × D | 4 | 0.34 | 0.981 | 1.16 | 0.31 | 1.91 | 0.033 * |
| A × B × C × D | 24 | 0.46 | 0.988 | 0.84 | 0.68 | 0.71 | 0.838 |
| Error | 293 | | | | | | |

Significance level: *p* < 0.05 *, *p* < 0.0001 **.

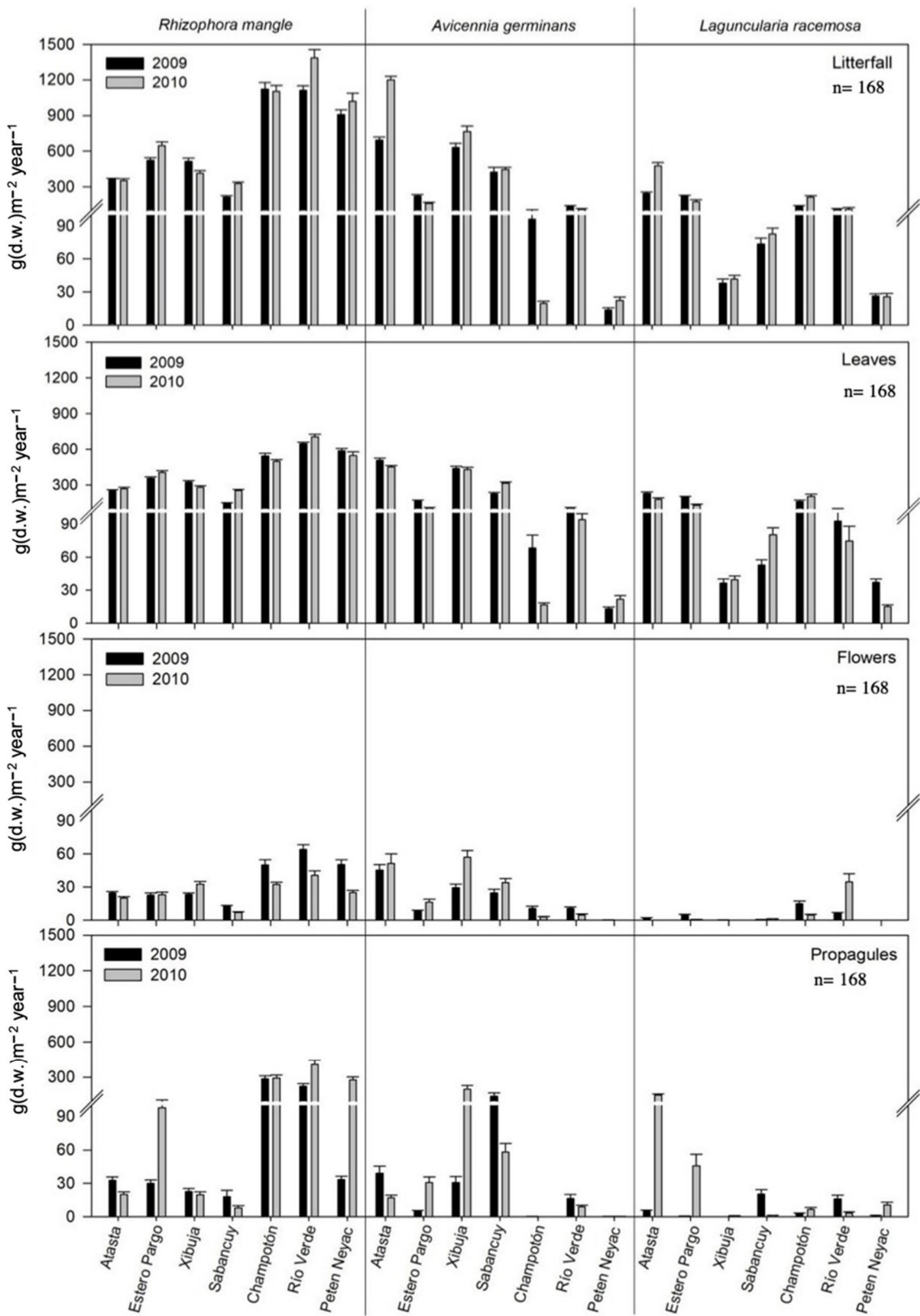

**Figure 7.** Litterfall and its components for each species (±SE = standard error), in the seven mangrove forests studied along the coast of Campeche for 2009 and 2010.

**Table 6.** Results of a three-factorial repeated-measurement ANOVA on the effects of site (Peten Neyac, Río Verde, Champotón, Sabancuy, Xibuja, Estero Pargo and Atasta), year (2009 and 2010), season (dry, rainy and Nortes) and species (*Rhizophora mangle*, *Avicennia germinans* and *Laguncularia racemosa*) on the salinity and redox potential along the Campeche coast, with a significance level of $p < 0.05$.

| Factor | df | Salinity (PSU) | | Redox Potential (mV) | |
|---|---|---|---|---|---|
| | | F | *p* | F | *p* |
| A (Site) | 6 | 113.75 | <0.0001 ** | 17.89 | <0.0001 ** |
| B (Year) | 1 | 4.24 | <0.040 * | 217.34 | <0.0001 ** |
| C (Season) | 2 | 13.47 | <0.0001 ** | 18.02 | <0.0001 ** |
| A × B | 6 | 1.66 | 0.131 | 3.21 | <0.004 * |
| A × C | 12 | 8.61 | <0.0001 ** | 4.98 | <0.0001 ** |
| B × C | 2 | 12.95 | <0.0001 ** | 36.74 | <0.0001 ** |
| A × B × C | 12 | 6.76 | <0.0001 ** | 7.09 | <0.0001 ** |
| Error | 360 | | | | |

Significance level: $p < 0.05$ *, $p < 0.0001$ **.

**Table 7.** Canonical correlation for pairs of linear combinations between the phenological production of the three mangrove species and the physicochemical parameters of the year 2009 with $p < 0.05$. Rm—*Rhizophora mangle*, Ag—*Avicennia germinans* and Lr—*Laguncularia racemosa*.

| | Canonical Variables | F1 | F2 |
|---|---|---|---|
| | **Canonical Correlation** | **0.754** | **0.544** |
| **Production Phenology** | | | |
| Leaves—*Ag* | | −0.461 | −0.503 |
| Flowers—*Ag* | | 0.276 | 0.433 |
| Propagules—*Ag* | | −0.211 | 0.897 |
| Litterfall—*Ag* | | 0.386 | 0.050 |
| Leaves—*Rm* | | −0.634 | −0.369 |
| Flowers—*Rm* | | −0.279 | 0.237 |
| Propagules—*Rm* | | −0.711 | 0.194 |
| Litterfall—*Rm* | | 0.499 | −0.204 |
| Leaves—*Lr* | | 0.211 | 0.036 |
| Flowers—*Lr* | | 0.234 | 0.266 |
| Propagules—*Lr* | | 0.315 | 0.041 |
| Litterfall—*Lr* | | −0.761 | 0.097 |
| Physicochemical parameters | | | |
| Salinity (‰) | | 0.994 | 0.287 |
| Redox potential (mV) | | 0.221 | −0.504 |
| Precipitation (mm) | | −0.093 | 0.909 |

*3.6. Pore-Water Chemistry*

The average salinity in 2009 (41.5 ± 13.9 PSU) was significantly higher (F = 4.23, $p < 0.040$) than 2010 (37.3 ± 16.9 PSU), although mesohaline–euhaline conditions were observed in both years: from 37.7 ± 9.5 PSU during the rainy season to 42.9 ± 16.1 PSU during the drought season.

Salinity was significantly different between the rainy and dry seasons in 2009 and 2010 (F = 13.47, $p < 0.0001$, Table 6, Figure 8). Furthermore, there was a significant year–season interaction (F = 12.95, $p < 0.0001$) (Table 6, Figure 8). In addition, the salinity shifted significantly between sites (Fisher's test: F = 113.75, $p < 0.0001$). Thus, salinity in the Atasta and Sabancuy sites in LT was significantly different compared with the Río Verde and

Peten Neyac sites in RBP ($p < 0.05$) (Table S3, Figure 8). Furthermore, salinity in Río Verde and Peten Neyac (RBP) was significantly different from that in CR ($p < 0.05$) (Table S3 and Figure 8). A significant inverse correlation was observed between salinity and precipitation ($r = −0.30$, $p < 0.002$).

In 2009, oxic to hypoxic and hypoxic to anoxic conditions (−245.6 ± 95.6 mV) were observed; oxic conditions were most prevalent in 2010 (−105.9 ± 36 mV) (Figure 8). Redox potential varied significantly between years ($F = 217.33$, $p < 0.0001$). Moreover, the year–season interaction was also significant ($F = 36.74$, $p < 0.0001$) (Table 6 and Figure 8).

In 2009, the pore-water redox potential indicated that conditions were more hypoxic to anoxic in RBP (−317.4 ± 31.2 mV) than in LT (−214.1 ± 97.2 mV). Conversely, in 2010, oxic conditions were observed in both RBP and LT (−88.8 ± 124.6 and −87.6 ± 89.9 mV, respectively, Figure 8).

The redox potential was significantly different between all pairs of seasons (dry vs. Nortes, dry vs. rainy and rainy vs. Nortes) ($F = 18.02$, $p < 0.0001$) (Table 6 and Figure 8). Fisher's LSD post-hoc tests following the nested ANOVA revealed significant differences between seasons and pore water redox potential of between 2009 and 2010 ($p < 0.05$) (Table S3 and Figure 8).

A significant difference in the redox potential was detected between varied significantly between sites ($F = 17.89$, $p < 0.0001$). Fisher's LSD post-hoc tests following the nested ANOVA revealed significant differences between Atasta, Xibuja and Sabancuy in LT vs. Río Verde and Peten Neyac in RBP ($p < 0.05$; Table 3) and between Atasta and CR ($p < 0.05$) (Table S3 and Figure 8).

A significant inverse correlation between salinity and redox potential was observed in all the forests in 2009 ($r = −0.40$, $p < 0.001$).

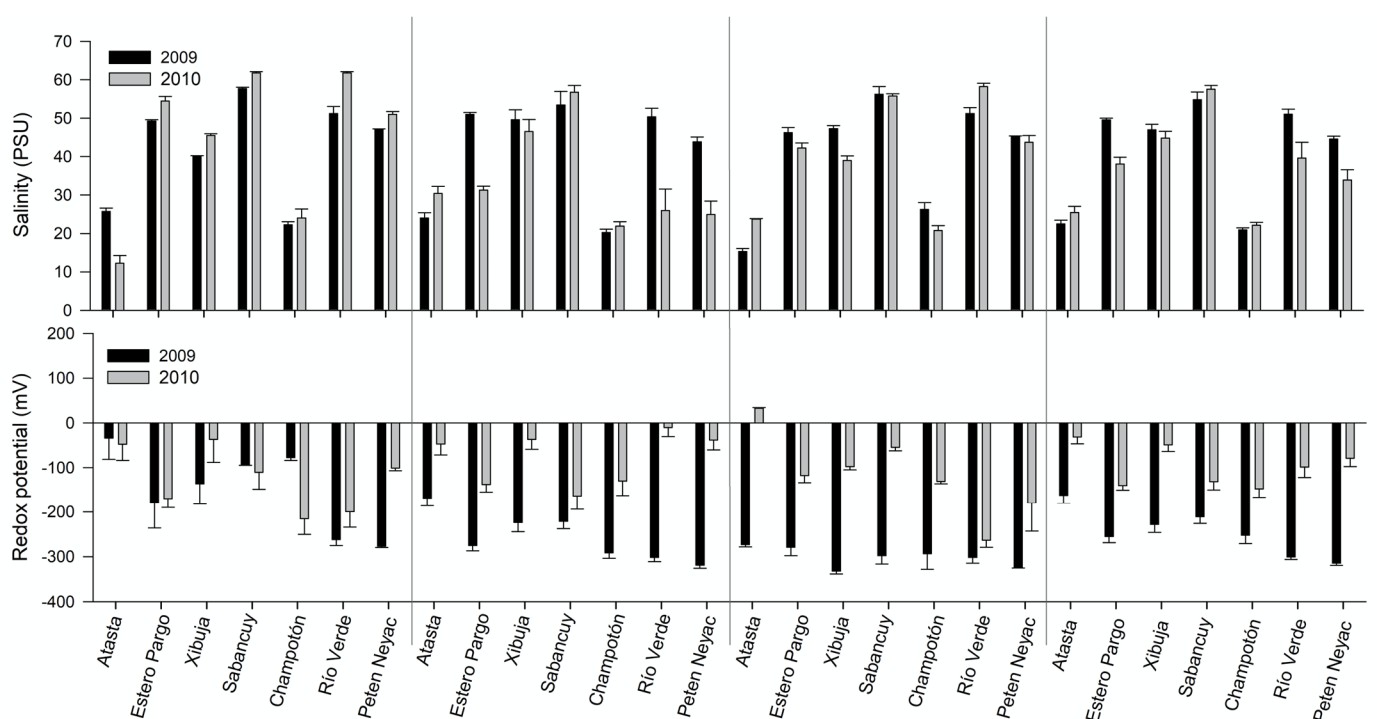

**Figure 8.** Pore-water salinity and redox potential (±SE = standard error) in seven mangrove forests along the coast of Campeche, Mexico.

### 3.7. Correlation between Pore-Water Chemistry and Reproductive Phenology by Species

During the severe drought in 2009, the first linear combination canonical correlation (F1) (Table 7 and Figure S1) showed that salinity correlated directly with litterfall production and inversely with leaves and propagules in *R. mangle*. Likewise, *L. racemosa* showed

an inverse relationship between litterfall production and salinity. For the second canonical correlation (F2) of the year 2009, the production of propagules and flowers of *A. germinans* stood out, and a direct relationship with precipitation and an inverse relationship with the redox potential was shown (Table 7).

In 2010, with average precipitation, according to the first linear combination canonical correlation (F1), the production of *R. mangle* flowers was directly correlated with the redox potential and precipitation, and inversely with the production of leaves of *R. mangle*. For the second canonical correlation (F2), a direct relationship between the production of *R. mangle* leaves with salinity is shown. (Table 8). Likewise, in general terms, the flower and leaf production of *R. mangle* is influenced by an increase in precipitation and a decrease in salinity (Table 8 and Figure S2).

**Table 8.** Canonical correlation between three mangrove species' phenological production and physicochemical parameters for 2010, with $p < 0.05$. Rm—*Rhizophora mangle*, Ag— *Avicennia germinans* and Lr—*Laguncularia racemosa*.

| | Canonical Variables | F1 | F2 |
|---|---|---|---|
| | **Canonical Correlation** | **0.683** | **0.511** |
| **Production Phenology** | | | |
| Leaves—Ag | | −0.260 | 0.121 |
| Flowers—Ag | | −0.447 | 0.200 |
| Propagules—Ag | | −0.132 | 0.152 |
| Litterfall—Ag | | −0.192 | 0.080 |
| Leaves—Rm | | 4.771 | 6.671 |
| Flowers—Rm | | −5.849 | −1.671 |
| Propagules—Rm | | 1.796 | −4.609 |
| Litterfall—Rm | | −0.089 | −0.229 |
| Leaves—Lr | | −0.785 | −0.527 |
| Flowers—Lr | | −0.106 | −0.190 |
| Propagules—Lr | | −0.319 | −0.631 |
| Litterfall—Lr | | 0.534 | 0.150 |
| Physicochemical parameters | | | |
| Salinity (‰) | | 0.231 | 0.883 |
| Redox potential (mV) | | −0.567 | 0.622 |
| Precipitation (mm) | | −0.789 | 0.015 |

## 4. Discussion

The effects of climate change are becoming increasingly evident; therefore, there is a need to understand the influence of global warming on environmental variables and the way this affects mangroves as precipitation is considered key to maintaining their productivity, diversity and distribution [44]. Sippo et al. [45] indicated that approximately 70% of publications report that the degradation and loss of mangroves are attributed to the high intensity of hydrometeorological events, such as tropical cyclones and extreme changes in climate.

### 4.1. Temporal and Spatial Variations in Pore Water Chemistry along the Coast of Campeche

Severe drought occurred in 2009, mainly in the rainy and dry seasons, which led to an increase in salinity and a decrease in the redox potential in the pore water. The most significant drought was recorded in LT (south), mainly in Atasta and Estero Pargo, during the rainy season in 2009, although two important water systems (Palizada River and the Candelaria River) constantly provide water to this area [46]. RBP (northern) is characterized by oxic to hypoxic and hypoxic to anoxic (<−300 mV) conditions, whereas in LT (southern), oxic–hypoxic conditions are prevalent. In general, inverse correlation between

salinity and precipitation (r = −0.30, *p* < 0.001) and between salinity and redox potential (r = −0.40, *p* < 0.001) in the seven mangrove forests was observed. Similarly, Saravanakumar et al. [12] mentioned that variation in dissolved oxygen concentrations in pore water depends on salinity concentration and temperature, which were inversely correlated. In response to the reduction in freshwater input into the wetland, the dissolved oxygen in the pore water decreases, increasing bacterial respiration demand and organic matter degradation, among other biogeochemical processes [47]. In addition, Houghton et al. [13] stated that when precipitation decreases, there is entry of fresh surface and interstitial water and soil salinity increases, which alters the average salinity in that season and year.

Previous studies have reported that high salinity and low freshwater inputs from precipitation, groundwater or river flows can lead to mangrove dieback, particularly in areas of restricted tidal flows or droughts [14].

The greater drought conditions and variation in rainfall between the 2009 and 2010 seasons in the seven forests located along the Campeche coast might be the result of climatic variation at the global and local levels. Microclimatic changes reduce the flow of rivers and increase the variability in their hydrological regimes, involving episodes of drought and changes in river systems [48]. The drought of 2009 was the most severe drought in Mexico in the past 50 years [49]. The results of this study may be related to the effects of El Niño, reported in Mexico in 2009 by the National Oceanic and Atmospheric Administration (NOAA) (2016), according to ONI [50]. In the last 20 years, El Niño was present during 1997–1998, 2002–2003 and 2009–2010.

In the study area, between 2009 and 2010, no extraordinary winds were identified at the height where baskets were installed to collect litterfall (1.5 m above ground), the wind speed is 50–70% lower than the height at which the wind speed is measured 10 m above ground (Figure S3). However, authors such as Shamim et al. [51], report a positive relationship between wind and litterfall collection, when there are nearby meteorological stations. For this study, we used a database with a resolution of 0.25 degrees. Therefore, it was not possible to identify a clear trend.

### 4.2. Responses of Mangroves to Variability in Pore-Water Chemistry at the Inter- and Intraspecific Levels

Forest structure and litterfall were correlated with the heterogeneity of environmental conditions along the Campeche coast. Forest attributes were higher in LT (southern) than in RBP (northern). The tallest mangrove trees (>15 m) were recorded at Atasta and Estero Pargo in LT, where *Avicennia germinans* is dominant and *Rhizophora mangle* was dominant at RBP. In contrast, *Laguncularia racemosa* was observed in all seven mangrove forests but was not dominant.

The forest structure, water and soil physicochemical variation conditions recorded in study sites were associated with nutrient contributions, salinity, dissolved oxygen, differences in precipitation and tides (period and frequency of flooding), intensity of evaporation and river presence or absence.

The mangrove forests in the southwest area of Laguna de Terminos receive nutrient inputs from the San Pedro, San Pablo and Palizada rivers [52], which lead to tree growth heights of 15 m.

On the other hand, the Laguna de Terminos eastern area does not present nutrients from any river to mangrove forests, causing a higher concentration of salinity, lower concentration of oxygen and less availability of nutrients during the rainy season, thus generating greater hydro stress, greater need to capture atmospheric oxygen and greater assimilation of ammonium as a source of nitrogenous nutrients for mangrove trees [53]. Numerous studies have shown that hydrological conditions and pore-water salinity determine the structure and diversity of mangroves in Veracruz [36,54,55]. Kamruzzaman et al. [56] found positive relationships ($R^2$ = 0.79) between the height and diameter of trees and nutrient inputs and litterfall in the mangrove forests of Sundarbans, Bangladesh. Similar relationships were observed by [23] in mangrove forests containing *R. mangle* ($R^2$ =

0.91) and *A. germinans* ($R^2 = 0.92$) in southern México. According to Agraz-Hernández et al. [21], mesohaline and oxic pore-water conditions enhance forest attributes in Atasta and Pom-Atasta mangroves, which are the most well-developed forests of LT (as well as in the state of Campeche) and are dominated by *A. germinans*.

Litterfall along the Campeche coast was significantly lower in 2009 than in 2010, which corresponded to a precipitation deficit in 2009. As a result, litterfall decreased by 18.2% in LT and 15.8% in RBP; the decrease in litterfall in the rainy season was also pronounced (35.8%). Leaf litter increased in the three mangrove species, mainly in *A. germinans*, when the salinity increased and the dissolved oxygen in the pore water decreased. Litterfall increased from the southwest (LT) to the northwest (RBP). The greatest litterfall ($1718.3 \pm 647$ g·m$^{-2}$·year$^{-1}$) was observed in LT, which receives constant terrigenous nutrients via runoff from the San Pedro, San Pablo and Palizada Rivers [52]. In contrast, the lowest litterfall ($1465 \pm 286$ g·m$^{-2}$·year$^{-1}$) was observed in RBP (Río Verde), which receives low superficial freshwater flow. In addition, this site is characterized by karstic soils, underground rivers and low nutrients and is affected by seawater to a greater degree [57]. In this regard, Osland et al. [58] mentions that the structure, function and distribution of mangrove species are closely influenced by climate extreme changes.

The maximum litterfall was from *A. germinans* in LT and from *R. mangle* in RBP. This indicates that mangroves exhibit high resilience in response to the wide variation in environmental conditions in which they develop [59]. The different responses recorded in litter production and phenological reproduction among the three mangrove species relating to the severe drought conditions in this study can be mainly attributed to differences in the tolerance intervals to salinity and anoxia among the three species [60].

*Rhizophora mangle* has ultrafiltration mechanisms and does not have desalination glands; it can only accumulate salt in the leaf vacuoles, making it vulnerable when salinity is greater than 35 UPS [61]. The increase in leaf litter is associated with an increase in salinity and temperature because increases in the high energy cost preserve photosynthetic tissue [62,63]. This species is distributed in sites with a higher frequency and intensity of flooding and higher availability of dissolved oxygen in interstitial water compared with coexisting mangrove species [64,65].

In contrast, *A. germinans* can excrete, exclude and accumulate dissolved salts in water, which permits this species to tolerate salinities from 25 to 120 UPS [66]. This species has pneumatophores and lenticels, which allow it to capture atmospheric oxygen and tolerate increased residence times under high water, drought and hypoxic conditions, including short periods of anoxia. As salinity increases and the redox potential of interstitial water decreases, leaf litter increases and biomass decreases, thus maintaining vital functions and reducing the energy cost of maintaining photosynthetic tissue [64,67,68]. *Avicennia germinans* is the mangrove species most resistant to extreme salinity conditions, hypoxia and prolonged drought [69].

In *L. racemosa*, lower litterfall production is observed under severe drought conditions as the supply of fresh water and nutrients decreases and salinity increases because this species depends on the hydroperiod and phosphorus supply [40,70]. Our results are similar to those of Lugo and Snedaker [71], showing that leaf litter is high under long periods of extreme drought. Consequently, the capacity to capture and store atmospheric carbon in arboreal biomass and soil for long periods is modified [4] and can contribute to 10% of GHG (greenhouse gas) emissions globally [5]. Psuty et al. [72] stated that *A. germinans* only grows in sites with scarce overflows and nutrients, whereas *L. racemosa* is frequently found in overflowing terrains with high nutrient contents. Thus, *R. mangle* may be considered an indicator of interannual and seasonal variation in precipitation because this species can tolerate a broader range of environmental conditions relating to ground and surface water. The mangrove forest bordering the RC channel was riparian, characterized by mesohaline conditions and a constant supply of nutrients and dominated by *R. mangle*. The presence of *L. racemosa* stemmed from the lower surface flooding and the presence of nutrients in the water due to agricultural activities in the area. Therefore, the

phenological results can be used as indicators of global environmental change, as described by the U.S. [73].

Reproductive phenology differed significantly between seasons and species in 2009 due to the precipitation deficit (56%) during the rainy season, mainly in southwestern (LT) Campeche. Low propagule and flower production was observed under low-precipitation and high-salinity conditions, even in extraordinary and common climatic events such as droughts. Propagule and flower production was decreased by 77.3% and 53.5%, respectively, in 2009 compared with 2010. In this year, propagules decreased by 60.7%, 22.2% and 67.7% during the rainy, Nortes and dry seasons, respectively. Pore-water redox potential (<−300 mV) was significantly positively correlated with annual flower and propagule production. The highest variation in flower and propagule production was recorded in LT, where *A. germinans* dominates because flowering is more sensitive to climatic variation in this species; by contrast, in *R. mangle*, propagules are more sensitive to salinity and pore-water redox potential [74]. Friess et al. [75] mentions that a decrease in rainfall reduces photosynthesis, productivity and reproduction of mangroves.

Rodríguez-Ramírez et al. [76] indicated that freshwater surplus is optimal for the reproduction of mangrove species. Under high salinity stress, *Rhizophora* requires energy-demanding metabolic processes and thus greater resources for the synthesis of osmolytes to absorb the water, thereby leading to a reduction in the photosynthetic rate, flowering, or bud maturation. These processes depend on the number of resources available after water and salinity regulation [67,77]. Several studies suggest that the reproductive phenology of *R. mangle* and *R. mucroniata* is affected by environmental factors; precipitation is a key factor in the flowering and propagule production of these species and is strongly associated with the energy balance [7,78].

Other studies have indicated that *R. mangle* reproduces throughout the year, but its production is higher in the rainy season when salinity is relatively low and nutrient availability is relatively high [23,54,79]. For this reason, Agraz-Hernández et al. [21] detected a strong positive correlation of propagule production with precipitation ($R^2$ = 0.60, $p <$ 0.0061) and redox potential values ($R^2$ = 0.53, $p < 0.05$) in a monospecific mangrove forest of *R. mangle* in LT. *A. germinans* only reproduces during the rainy season. Modifications to the reproductive behavior of *A. germinans* under severe drought conditions (2009) stem from water stress caused by decreasing precipitation and increasing salinity, which affects the vital functions of plants. Water stress affects the reproductive phase, especially pollination and propagule development [80,81], indicating that *A. germinans* and *A. bicolor* undergo changes in flower production under water stress. In *L. racemosa*, photosynthetic activity, growth and reproduction are increased under low water stress. This stems from the efficiency with which they can use nitrogen and phosphorus [82], which allows them to respond rapidly to periods in which water is more abundant. In this way, *L. racemosa* grows and reproduces during periods when water is most abundant [70,83].

Das et al. [84] and Srikanth et al. [85] stated that abiotic stresses include excess salinity, an excess of metal ions and waterlogging conditions contributing to an anaerobic environment. Changes in the precipitation regime can lead to climatic variation that affects the growth and survival of propagules in mangrove forests [86], even causing drought scenarios that cause the death of mangroves [82]. Predictive models from the IPCC (2001) of mangrove forests of Central America and Australia indicate that precipitation may decrease in certain areas due to changes in the precipitation regime.

*4.3. Species Show the Greatest Adaptation under Severe Drought*

Mangrove ecosystems along the Campeche coast are characterized by differences in physiognomy and dominant species. *Avicennia germinans* and *L. racemosa* showed greater adaptation than *R. mangle* throughout the state of Campeche under severe drought conditions. Mangrove communities are particularly vulnerable to changes in precipitation because the hydroperiod and pore-water chemistry are directly modified by climatic variation or global anthropogenic change (climate change) [44,87]. A small decrease in

precipitation causes a reduction in the contribution of freshwater, which makes the soil more saline; even slight modifications in hydrological conditions can lead to substantial changes in the composition, species richness, net primary production, productivity of biota, growth and survival [88]. Decreases in precipitation can also affect carbon storage, which is one of the most important ecosystem services provided by mangroves because they fix atmospheric $CO_2$ and transform it through photosynthesis in biomass [89]. Climate change may have its most pronounced effect on coastal wetlands by altering hydrological regimes. The frequency and duration of the tides, rivers and runoff in the region and flooded areas affect pore-water chemistry in mangrove habitats. Increases in drought frequency lead to decreases in reproduction, adverse structural changes (reductions in area and canopy) and increases in the vulnerability of mangroves to other natural and anthropogenic stresses [90,91].

## 5. Conclusions

The composition and abundance of species in the mangrove forests are related to hydrological conditions. The hydrological variations that occurred in 2009 caused significant changes in the physiological indicators studied in populations of mangroves throughout the study area. These changes were more marked in the Laguna de Terminos where the drought was classified as severe.

This study showed that the hydrological changes due to the drought year (2009) in the mangrove forests of the state of Campeche caused significant changes in the hydrochemical conditions in the area, significantly affecting the reproductive phenology, specifically in propagule production in *A. germinans* in the Laguna de Terminos, in *R. mangle* in the Los Petenes Biosphere Reserve and in the production of flowers in *A. germinans*. These findings confirm that extreme drought events can reach thresholds that cause physiological stress in *A. germinans*, *L. racemosa* and *R. mangle* Additionally, it confirms the vulnerability of the mangrove ecosystem in the current and future scenarios of natural climate variations or global anthropogenic change (climate change), from local and regional rainfall variations. Therefore, if climate changes continue, the mangroves will be exposed to variations outside the tolerance intervals in physical and chemical, interstitial water and soil conditions, generating ecosystems of low productivity and diversity, which is endangering the integrity of mangrove.

Finally, the results of this study showed that *A. germinans* and *L. racemosa* showed lower values than *R. mangle* throughout the state of Campeche under severe drought conditions. The magnitude and intensity of the impacts caused by climatic variation will generate heterogeneous environmental changes and therefore, the mangrove species will present different responses depending on their location. Likewise, the results of this research provide a baseline that could be used to enhance our understanding of the ecological functioning and management of mangrove communities with important applications. It is necessary to consider the effects induced by environmental changes generated from climate change and the planning of restoration initiatives, such as understanding the tolerance thresholds that the species of mangrove have before ecosystem death happens. We depend on understanding the reproductive phenology of mangrove species to determine whether propagules are available in various environmental conditions. As a result, we think that environmental restoration is a crucial strategy for restoring mangrove cover and, in turn, the ecological services it provides. In addition, prediction models could facilitate the development of mitigation strategies in the face of environmental changes in mangroves stemming from climate change, according to the geomorphological characteristics, physiographic and climatic types and anthropogenic pressures in mangroves.

## 6. Implications for Conservation

The phenological data in this study are crucial for understanding the influence of environmental changes on phenological patterns of mangrove reproduction. We propose that phenological behavior analysis could be used as a bioindicator with high sensitivity

to climatic variation. Mangroves are sessile throughout their life and survival in their habitat requires phenological resilience. The results of this research provide a baseline that could be used to enhance our understanding of the ecological functioning and management of mangrove communities and the species themselves. Thus, prediction models could facilitate the development of mitigation strategies in the face of environmental changes in mangroves stemming from climate change, according to the geomorphological characteristics, physiographic and climatic types and anthropogenic pressures in mangroves. In fact, all of this information has already been applied successfully in restoration projects for mangroves on the coast of Campeche, where information on drought events has been used in hydrological models to develop innovative rehabilitation technology (CONACyT/FOMIX/126430; CONABIO/FN010; CONABIO/KN001). This technology has been applied to mangrove ecosystem restoration programs in other countries (Costa Rica and Benin, Africa) (CONAFOR2014–2017; CONAFOR2017–2022; FFEM/NEOTROPICA/SINAC/MINAE/UAC2018–2021; CORDE/UAC2018–2021), and in all cases, this technology has promoted the recovery of ecosystem services.

## 7. Recommendations

Based on the results of this research, it is recommended that similar long-term studies should be carried out to record different effects on the structure and function of mangroves in the face of variation in the intensities and magnitudes of drought and, consequently, to understand how mangrove species establish the capacity for adaptation and resilience to events generated by climatic variation over time. In addition to the data obtained, it will be possible to feed a model that will allow for the prediction of the threats mangroves will encounter in the face of the effects produced by climate change, thus assisting decision-makers to apply faster and more effective mitigation measures based on the needs of each ecosystem and its dominant species. It is still unclear as to what extent mangroves may have the capacity to recover after an intense drought.

**Supplementary Materials:** The following supporting information can be downloaded at: https://www.mdpi.com/article/10.3390/d14080668/s1, Figure S1: Bi-plot correlation between three mangrove species phenological production and physicochemical parameters (canonical variables) in 2009. Figure S2: Bi-plot correlation between three mangrove species phenological production and physicochemical parameters (canonical variables) in 2010. Figure S3. Wind 10 m above ground, 2009–2010, ERA 5 Database along Campeche coast. (a) Carmen City, (b) Champoton, and (c) Los Petenes Biosphere Reserve. Table S1: Structural metrics computed for the structural analysis (adapted from Cintron et al. [33] and Akodekou [34]). Table S2: Summary of Fisher's Least Significant Difference post hoc tests conducted for the nested ANOVA on litter fall production and components in comparison with sites and seasons. *Ag*: *Avicennia germinans*; *Lr*: *Laguncularia racemosa*; *Rm*: *Rhizophora mangle*. Peten Neyac, Río Verde, Champotón, Sabancuy, Xibuja, Estero Pargo and Atasta, with a significance level of $p < 0.05$. Table S3: Comparison of Fisher's Least Significant Difference post-hoc tests conducted for the nested ANOVA on potential redox, salinity and precipitation in comparison with sites and seasons, with a significance level of $p < 0.05$.

**Author Contributions:** C.M.A.-H., C.A.C.-K., R.M.-S. and R.A.P.-B. conceived of the study, participated in its design and coordination and helped to draft the manuscript. C.M.A.-H., J.O.-S. and H.G.M. collected and analyzed the data. C.M.A.-H., C.A.C.-K., R.M.-S., R.d.R.R. and G.P.V. helped with analyses and interpreted data. C.M.A.-H., C.A.C.-K. and R.M.-S. wrote, reviewed and edited the manuscript. All authors have read and agreed to the published version of the manuscript.

**Funding:** This research was funded by The Biodiversity Federal Agency (CONABIO; Comisión Nacional para el Conocimiento de la Biodiversidad en México) Grant Numbers FN010 and KN001.

**Institutional Review Board Statement:** SEMARNAT No.SGPA/DGVS08059/17.

**Data Availability Statement:** Data is contained within the article.

**Acknowledgments:** We thank Gerardo Martinez-Kumul, Gilberto Martinez-Muñoz and Kenia Pahola Conde Medina for their help during field sampling.

**Conflicts of Interest:** The authors declare no conflict of interest.

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
