# Peer review of "Pore Water Chemical Variability and Its Effect on Phenological Production in Three Mangrove Species under Drought Conditions in Southeastern Mexico"

_diversity, doi:10.3390/d14080668_

Round 1

Reviewer 1 Report

This is a significant study and it provides insights into effects of climate variability on mangrove species. 

However, there are some corrections to be made. First presentation of results, which appears to be too detailed and may only interest science and miss out on the policy and decision making. Some of the results, especially statistics could be presented as supplementary materials and referred to in the main article. This is to strike a balance for wide readership including the policy makers who are interested more in synthesized results. 

Secondly, the implications of results need to be given more emphasize and some recommendations be given.

Other corrections relate to the methodology and general write up

1. Clearly indicate the study site and connect it with the north-south description. The authors described the north-south divide, but did not indicate where the study sites fit, and failed to justify how the N-S description was necessary in relation to the study site.

2. Consistency in citation format - e.g., in line 57, the authors ought to indicate reference 10 followed by a number (10) in parenthesis, since ref 10 is the subject of the sentence, as in the case of other references which are subject matter, e.g., in line 69

3. Rephrase sentence in line 77-78, seems ambiguous.

4. Line 141 - what position is diameter measured for stems with prop (stilt) roots? Is it 30 cm above the lowest root or the highest root? Conventionally, the diameter of Rhizophora stems is measured 30 cm above the highest prop-root.

5. Line 229 - capitalize the first letter at the beginning of the sentence

6. Line 643 - revise the sentence - it seems some sections incomplete "influence of environmental........"

Author Response

See attachment for details

Reviewer 2 Report

General comments

My first impressions of the title are not fully supported by the results, so I recommend changing it. The findings were more focused on drought and litterfall patterns than on their relationships 8for example how salinity and redox potential directly influence litterfall should be addressed. In the case of introductions, question three is not fully supported. Thus, a little background information is highly recommended. Methods need more explanation in general, and statistical tests need more attention in particular. For example, the bi-plot analysis description is missing in the text but is shown in the results. Question 3 is poorly focused on the results sections as well. I would not add many figures in the text that are not highly related to the questions. I am also wondering how authors avoid the effects of wind on litterfall production. Rainfall and wind combine to influence litterfall (Ahmed and Kamruzzaman, 2021). As a result, if possible, I would like to request the addition of a wind effect or to discuss it as a limitation. Besides, the dataset used in this study were collected in 2009 and 2010. Why are the authors presenting an old dataset? I want to know the current state and conditions of forest and meteorological divergence in this area from 2009–2010. I think the discussion would be interesting to read if it followed the questions. I would restructure it based on the questions. Conclusions need to be restructured with the main messages and limitations of the study. Few words need to be revised, for example, 'registered', "significantly different lower’, etc. are discouraged from being used and changed to more understandable words. Numerous grammatical mistakes and typos are found, recommend checking with a native speaker. I hope the following specific comments will be helpful for the authors to improve their manuscript.

Specific comments

L: 135, y? what does it mean?

L: 138. How authors confirmed, that only two quadrats represent all species in particular sites?

L. 140. DBH abbreviated form required.

L. 140. better to write measured rather than determined.

L: 142, sentence incomplete. ‘By using….’. I think the authors wanted to write about height measurement.

L148: I would not add Table 2 in the main text instead in supplementary.  I doubt the height calculation, why authors did not follow the same as the mean D calculation. Besides, how the authors measured H is not clear 

L: 154, 50*50 m. litterfall baskets? It seems litterfall baskets are bigger than plot sizes.

L:155-156. How do you place baskets? Randomly?

L: height from forest floor? Litter and trap are susceptible to strong winds which can strongly influence litterfall, is there any cyclonic events that happened during the study periods?

L: 158. How authors defined the constant weight? It’s always critical to do that. Generally, 48 hours is recommended. Need an explanation on this.

L. 160, how authors separated unwanted things, is not clear to me. Add, please. 

L:161. Company name required.

L: 165. It’s not clear to me, whether the authors collected litterfall monthly or daily. In the previous discussion, it seems monthly or I missed something.

L 117, 165, 166: either ‘equation or Equation’, follow consistent patterns.

L 172: Were there any tidal effects during the study? If so, how do they avoid it? When did pore salinity measure like when tidal water was established? In addition, rainfall strongly influences these parameters, how did they overcome this influence?

L 179: again methodological approaches missing (random or systematic)?

L 180. Finally found. Thanks

L 182. in situ salinity was measured?

L193: Two-way repeated measures ANOVA? Analysis of variance? Three factorial?

3. Results and discussion? I can see some comparative results are presented

L207-2010. More fit methods section.

L. 259. Remove space

L259. Change to ‘the’

L274. Better to write ‘significantly lower’.

L275. Not clear. Increase clarity.

L281. Fig. 6 does not show any percentage value, so either % in Fig. 6 or removed % value from the text.

L342. Explain a bit what does mesohaline-euhaline condition means. Medium to high?

L449. References required.

L-476. did the authors analyze nutrients or took it from literature? Need reference

L553- change ‘motions’ to mentioned

L-617, ‘confirms’

Ahmed, S., Kamruzzaman, M., 2021. Species-specific biomass and carbon flux in Sundarbans mangrove forest, Bangladesh: Response to stand and weather variables. Biomass and Bioenergy 153, 106215. 

Author Response

See attachment for details

Round 2

Reviewer 2 Report

I appreciated the author's effort in trying to solve the raised issues. However, the manuscript can be accepted after explaining why the wind effect was ignored. Although the authors nicely explained the reasons, including such sentences in the main text would make it more understandable to the readers. Therefore, they can use the following references: I also recommend adding the current comparative climate conditions to the supplementary so that readers can easily find the differences.

Ahmed, S., Kamruzzaman, M., 2021. Species-specific biomass and carbon flux in Sundarbans mangrove forest, Bangladesh: Response to stand and weather variables. Biomass and Bioenergy 153, 106215.

Author Response

See attachment for details
